# A Qualitative Analysis of Attrition in Parent–Child Interaction Therapy

**DOI:** 10.3390/ijerph192114341

**Published:** 2022-11-02

**Authors:** Amber Ufford, Tali Wigod, Joy Shen, Alec Miller, Lata McGinn

**Affiliations:** 1Private Practice, New York, NY 11215, USA; 2Cognitive and Behavioral Consultants, New York, NY 10025, USA

**Keywords:** parent–child interaction therapy, treatment attrition

## Abstract

Parent–child interaction therapy (PCIT) is one of the strongest evidence-based treatments for young children with behavior problems. Despite the efficacy of PCIT, many families fail to complete treatment, with attrition rates ranging from 30 to 69 percent. Preliminary research on attrition in PCIT treatment studies has linked maternal distress, negative verbal behavior (critical and sarcastic comments towards the child), lower socioeconomic status (SES), and fewer child major depressive disorder (MDD) diagnoses with premature termination from PCIT. However, more research is still needed to identify the range of reasons for treatment discontinuation. The purpose of the present study was to explore the range of reasons for premature termination from PCIT by conducting in-depth interviews with parents who discontinued PCIT using a qualitative design methodology. Results yielded eight themes, which were organized into three constructs: child-directed interaction (CDI) successes, difficulties with treatment, and the need for more clarity and orientation. Several existing treatment strategies that emerged from the data could be applied to PCIT to further enhance it and potentially reduce dropout (e.g., reconceptualizing dropout from PCIT, micro-orienting strategies used in other cognitive and behavioral therapies and dialectical behavior therapy). Understanding the reasons why parents drop out of PCIT and exploring different adaptations that can be made can further enhance this evidence-based treatment and increase its accessibility.

## 1. Introduction

Behavior problems in children are associated with a variety of immediate and long-term negative effects on children and their families. It is no surprise that disruptive behavior disorders (DBDs) are the most common reason for referral to mental health services among young children [1,2,3]. Young children with DBDs are more likely to experience peer rejection, academic difficulties, emotional and physical abuse, and other mental health problems. DBDs are also among the most powerful risk factors for subsequent delinquent behaviors in adolescence and adulthood, including interpersonal violence, substance abuse, and destruction of property [4], and for the development of severe psychopathology, such as antisocial personality disorder [5]. Children with DBDs account for a larger percentage of healthcare costs than children with chronic health conditions [6]. If left untreated, disruptive behaviors and related problems are often chronic and show a high degree of stability over time [7]. As such, early intervention programs for young children with DBDs are crucial. 

Fortunately, participation in effective parenting programs can bring relief to families and reduce the symptoms of DBDs to subclinical levels. Parent–child interaction therapy (PCIT) is an evidence-based treatment for young children between the ages of 2.5 and 7 who exhibit disruptive behavior disorders [8,9]. PCIT is based on the developmental theory that children need both warmth and limit setting to optimally develop [10]. This theory is reflected in the two treatment phases, child-directed interaction (CDI) and parent-directed interaction (PDI). Each phase begins with a didactic session during which parents are provided skills training. During each subsequent session, parents are provided with live feedback and coaching by therapists as they play and interact with their child. Families are considered to have met graduation criteria of PCIT after they reach expertise in the specific parenting skills taught during each phase, demonstrate confidence in managing their child’s behavior on their own, and report below-threshold levels of child behavior problems as measured by the Eyberg Child Behavior Inventory (ECBI; [11]). While such strict graduation criteria ensure that the greatest therapeutic gains are achieved, the criteria can be difficult for families to meet. 

Families who complete PCIT see significant improvements in child behavior problems [12]. PCIT results in increased positive parenting behaviors, such as praise and reflective listening and decreased negative verbal and physical behaviors toward the child during parent–child interactions. Children show increased compliance to parent directives and decreases in disruptive behaviors [13,14] to subclinical levels for disorders such as conduct disorder and oppositional defiant disorder [15,16]. Research shows that treatment gains generalize to treated children’s school behavior and to the behavior of untreated siblings [17,18,19] and lead to reduced stress in parents [20]. Further, improvements gained in PCIT are often maintained for years following treatment [21]. Overall, decades of research demonstrates that PCIT is an effective treatment for reducing disruptive behaviors in young children and relieving distress in the family. 

Despite the efficacy of PCIT, many families fail to complete treatment, with attrition rates ranging from 30 to 69 percent [22,23]. In PCIT, attrition is defined as discontinuing treatment at any point after attending the first treatment session and before meeting the treatment graduation criteria [21]. Preliminary research on attrition in PCIT treatment studies has linked maternal distress, negative verbal behavior (critical and sarcastic comments directed towards the child), lower socioeconomic status (SES), and fewer child major depressive disorder (MDD) diagnoses with premature termination from PCIT [24,25]. A few studies have also examined therapist behaviors that may impact premature termination. For example, Barnet et al. [25] found that coaching style, specifically the use of more responsive techniques and fewer skill drills, was associated with treatment completion. Additionally, Harwood and Eyberg [23] discovered that therapists of families who dropped out of treatment used significantly more supportive statements and fewer facilitative statements than therapists of families who completed treatment. Reasons identified by families for their premature termination include logistical problems (e.g., transportation, childcare), a belief that treatment did not progress quickly enough, frustration at the length of time spent on waitlists, and a general dislike of the treatment approach and techniques [15]. 

However, given the high attrition rates, and the consequences of dropping out of treatment for children, caregivers, and for society at large, more research is still needed to identify the range of reasons for treatment discontinuation. In their recent study examining families’ experiences in PCIT, Liebsack et al. [16] examined factors associated with attrition, such as therapy attitudes, expectations, caregiver commitment to treatment, and cultural competence of the therapist. Caregivers and therapists were also asked about whether families had completed PCIT or left prematurely. Results indicate that there was low caregiver–therapist agreement on treatment progress and completion, highlighting that therapists may not be aware or understand the reasons families terminate PCIT early. More research is clearly needed to investigate and fully understand the perspectives of caregivers who make the decision to drop out so that treatment can be better tailored to address and manage their concerns. 

While therapists have a limited ability to address the logistical barriers that can influence attrition rates, it is possible that therapists can address factors such as coaching style and parents’ perception of treatment, and possibly provide additional therapeutic or tangible resources to reduce parental stress. Yet, despite the reasons for premature termination identified thus far, only a few interventions have been implemented to combat this issue. In one such example, Fernandez and Eyberg [26] modified their PCIT protocol to specifically address maternal distress. PCIT therapists were required to allot a brief amount of time in each session to address parents’ personal concerns. While Fernandez and Eyberg did not specifically assess the impact of this modification on treatment attrition, their attrition rate was 36%, which is low compared to most evaluations of PCIT. In another example, Chaffin et al. [27] sought to improve retention and increase engagement in treatment through the incorporation of motivational interviewing (MI) techniques in PCIT with families referred by child welfare. In this study, benefits were primarily seen in families who reported low-to-moderate initial motivation. More recently, Quetsch et al. [28] evaluated the impact of incentives on homework completion, skill attainment, and attendance in Latinx families receiving PCIT. Families who received incentives demonstrated significantly lower no-show rates, though incentives had no impact on attrition or attainment of skills. Based on such findings, there is a need for additional studies to identify the reasons why families decide to drop out of treatment so that relevant strategies and interventions can be developed and incorporated to target those reasons and help children and parents remain in treatment. 

It is imperative that we engage parents in discussion to fully understand the reasons why families make the decision to drop out of PCIT so that we can better target strategies to reduce dropout and improve treatment outcome. The purpose of the present study was to explore the range of reasons for premature termination from PCIT by conducting in-depth interviews with parents who discontinued PCIT using a qualitative design methodology. By understanding why parents decide to leave treatment early, we hope to tailor intervention to better address their concerns and increase treatment engagement.

## 2. Materials and Methods

### 2.1. Participants 

Seven participants who received treatment at a fee-for-service clinical center were recruited for the present study. Participants were parents of children who had attended at least one session of PCIT and had dropped out of treatment before meeting graduation criteria. 

All parents identified as Caucasian (100%). Five families indicated that they had an annual household income of more than USD 300,000 (71.4%), and two families indicated their income was less than USD 80,000 (28.6%). Three families had two parents participate in treatment before dropping out, though only one parent of each family participated in the interview for the study. Three families had a total of three children in the family (42.9%), three had two children (42.9%), and one family only had the one identified child (14.3%). Parent employment included: financial analyst, travel agent, school psychologist, medical doctor, non-profit program director, business analyst, insurance salesman, TV producer, and homemaker. 

Parent-reported pretreatment intensity scores for the frequency of their child’s disruptive behavior on the ECBI ranged from 127 to 184, with a mean of 147.1 (SD = 24.6), suggesting that on average parents perceived their child’s behavior problems to be in the clinically significant range. Parent reported ECBI intensity scores at treatment dropout ranged from 68 to 186, with a mean of 120.6 (SD = 41.7), which is below the clinical cutoff of 131. Parent-reported pretreatment problem scores on the ECBI ranged from 7 to 29, with a mean of 17.1 (SD = 7.6), and post-treatment problem scores ranged from 3 to 29, with a mean of 12.3 (SD = 10.2). Six children were in the clinical range on the ECBI at pretreatment, while only three remained in this level at dropout. Children ranged in age from 5 to 7, with a mean age of 6.4 (SD = 0.78). Table 1 summarizes demographic information for the sample. 

Graduation criteria are defined as meeting goal or expert criteria in CDI, including mastery of the CDI skills (e.g., using 10 labeled praise, 10 behavioral descriptions, and 10 reflections, and less than 3 questions/commands/criticisms in a 5 min span), and expert criteria in PDI, which are defined as giving at least 75% effective commands and demonstrating at least 75% correct follow through with direct commands (e.g., following through with a labeled praise after the child obeys or a warning if the child disobeys, and if needed, correctly following through with the timeout sequence, including the timeout room if needed). Additionally, to graduate from PCIT, parents should demonstrate ECBI intensity scores of 114 or below and express confidence in managing their child’s behavior on their own.

### 2.2. Procedure

Potential participants were first identified by the PCIT clinical team. Team members notified researchers when a family dropped out of treatment before meeting graduation criteria. Once a case was identified as a “drop-out”, a PCIT research team member contacted the caregiver via email to assess interest in participating in the study. A PCIT research team member then followed up with the participant with an additional email and up to two phone calls to answer any questions about the study. If the participant indicated that they were interested, a PCIT research team member requested consent and demographic data. Afterwards, an interview was scheduled. Interviews were conducted over the course of a year (May 2020–May 2021).

The interview was semi-structured and consisted of 15 questions (see Figure 1). Interviews typically lasted 45 min, ranging in length from 30 to 60 min depending on the depth of the participants’ answers. The interviews were recorded and transcribed verbatim using a transcription software. Theoretical saturation was reached after interviewing seven parents, which is the criterium used in qualitative designs to determine that a sufficient number of participants have been interviewed in order to develop theoretical constructs [29].

### 2.3. Measures

*Demographic questionnaire.* The demographic questionnaire included questions regarding race/ethnicity, income levels, occupations, and number of children of the participants. 

*Eyberg Child Behavior Inventory (ECBI).* The ECBI [30] is a parent-reported inventory that measures conduct problems exhibited by children aged 2–16. It is a 36-item behavioral inventory that gathers data on the frequency of disruptive behaviors (intensity score) and whether they are considered a problem (problem score) on a 7-point Likert scale. A score on the intensity scale of 131 or more is considered a clinically significant indication of behavior problems. Previous research has demonstrated that the ECBI has good discriminant validity [31,32], concurrent validity [33,34], and high internal consistency [35].

### 2.4. Data Analysis

We used the grounded theory methodology developed by Auerbach and Silverstein [36] to code the current data. According to Auerbach and Silverstein [36], qualitative research involves analyzing and interpreting texts and/or interviews in order to discover meaningful patterns representing a particular group’s experience. More specifically, this grounded theory method allows researchers to generate hypotheses after evaluating the data, rather than the more traditional method of formulating hypotheses prior to data analyses.

Following the Auerbach and Silverstein [36] method, recorded transcripts were reviewed by two trained coders, who were research assistants. Relevant text, defined as any content related to the research topic and repeating ideas, which are similar words or phrases used by two or more participants to express the same idea, were identified by the coders. A coding meeting then occurred, allowing for discrepancies in the coding process to be identified. Next, the repeated ideas identified by both coders were individually categorized by common themes. Auerbach and Silverstein [36] define a theme as an implicit topic that organizes a group of repeating ideas. A second coding meeting then occurred to evaluate the inter-rater reliability between coders. After examining for potential discrepancies, the coders individually organized themes into abstract constructs. Finally, group members met with the current writers to organize the theoretical constructs into theoretical narratives, thereby providing a summary of the subjective experiences responsible for dropout rates of PCIT. 

## 3. Results

### 3.1. Dropout Rates

Over the course of six years, 56 families participated in PCIT at our clinic. Forty-eight percent of families did not meet the graduation criteria of PCIT (*n* = 27). Of these 27 families, 12.5% (*n* = 7) switched to a different treatment modality within the clinic (e.g., DBT-C, parent management training). Four of the twenty-seven families reported at termination that the reason for their dropout was due to the COVID-19 pandemic, explaining that it would be too difficult to continue treatment remotely from their home. In sum, a total of 18 families (32%) dropped out of treatment without switching to another modality or citing COVID-19 as their reason. Additionally, of the 56 families who participated in PCIT at CBC, 50% of them attended in-person therapy (*n* = 38), whereas 27% participated in internet-based PCIT (*n* = 15), and 5% switched to remote-based therapy at the start of the pandemic (*n* = 3). Figure 2 summarizes these results. 

Since previous studies examining dropout from PCIT have cited barriers to treatment (e.g., childcare and transportation) as reasons for premature termination, we were interested in assessing whether dropout rates changed post-March 2020, when our practice transitioned to internet-based PCIT. The dropout rate pre-March 2020, only including those who dropped out of our center (i.e., did not transition to another treatment modality) was 32.5% (*n* = 13), out of a sample of 40 families. When the same criteria were used in the post-March 2020 group, the dropout rate decreased to 31.3% (*n* = 5), out of a group of 16 families. 

### 3.2. Themes and Theoretical Constructs 

We identified eight major themes based on the reasons identified for premature termination from PCIT and, based on these themes, developed three theoretical constructs. The model was generated, in-line with Auerbach and Silverstein’s [36] grounded theory methodology, by first taking a bottom-up approach, which reflected parents’ subjective experiences of PCIT. We organized our themes into three constructs: CDI successes, difficulties with treatment, and the need for more clarity and orientation. A top-down approach was then used by applying existing treatment strategies related to our data, including reconceptualizing dropout from PCIT, micro-orienting strategies used in other cognitive and behavioral therapies, and dialectical behavior therapy (DBT; [37]), motivational interviewing (MI), and acceptance and value identification strategies used in acceptance and commitment therapy (ACT) to best understand parents’ experiences. The three theoretical constructs are listed below, with themes discussed under each construct. 

#### 3.2.1. Theoretical Construct 1: Positive Experiences in CDI That No Longer Necessitated Treatment

One theme identified was that parents had positive experiences in CDI, and often, as a result, no longer felt that treatment was needed. Parents noted that at the beginning of treatment, their “child really needed PCIT” and that they “really liked their clinician”. This suggests that at the beginning of treatment, they felt that PCIT could help them and was appropriate. However, some parents noted that as time went on and they moved to PDI, they believed: “We have reached a really amazing point right now” and “we don’t need to keep moving forward”. These parents reported feeling satisfied with progress during or immediately after CDI, and no longer felt the need to continue with treatment. 

#### 3.2.2. Theoretical Construct 2: Need for More Clarity and Orientation Upfront

This construct was supported by two broad themes. The first broad theme was the need for more information about the financial costs and burden of treatment upfront. Many parents reported on the burden PCIT placed on the family in terms of cost and time. Parents also stated that, if they had realized how long treatment would be at the outset, they would have felt more prepared for the overall cost and for the time commitment it would require. The second broad theme was the need for more orientation about treatment procedures at the onset of treatment. One parent stated, “…it’s not going to be like a miracle, problem solver despite all those efforts. So I think that should get a little bit of a disclaimer at the beginning”. Relatedly, parents also reported wanting more orientation about the rationale for both CDI and PDI, as some parents stated, “I didn’t understand how [CDI] was going to help us with his aggression,” and “I didn’t know why we had to do timeouts, it felt irrelevant for us. He would just sit there [in the timeout chair] and look at us, like ‘why are you doing this.’” Finally, parents also reported that they felt their kids needed more clarity and orientation about what was happening in treatment. One parent stated, “you need to have a transparent conversation with them about it, [PDI]”. Another parent stated about the timeout room “I think it was a little scary because the kids don’t know why they’re in there and they know why they’re in there, but they don’t know if they’re ever going to be allowed to just get out…”. Increasing clarity and providing additional orientation before and during treatment may help parents and children better understand the rationale for PCIT.

#### 3.2.3. Theoretical Construct 3: Difficulties and Challenges with PCIT 

This construct was supported by six broad themes including challenges with the structure of PCIT, difficulties with the pace and rigidity of the expert criteria, concerns with fit of PCIT, challenges with PDI, difficulties due to COVID-19 pandemic, and other factors.

Challenges with the Structure of PCIT. The first broad theme indicated that parents experienced challenges with the structure of PCIT. Parents described PCIT as being more difficult than expected. They experienced challenges using skills, both in session and at home. Parents also reported feeling that both phases of PCIT (i.e., CDI and PDI) felt unnatural and artificial. They reported that both the playtime and being coded felt unnatural. Parents also reported struggling to accurately complete the ECBI, saying that they often under-reported or filled it out the same way each week, as it became repetitive. Parents also reported not understanding the need to complete the ECBI each week, as their results were not always reviewed and the connection to the treatment was not made clear. Finally, parents also voiced concerns that PCIT sessions were not addressing misbehavior observed in the home, either because their kids were being “perfect angels” during the coding sessions, or because they did not understand how PCIT would ultimately address these recurring behaviors or did not have an opportunity to discuss these behaviors due to the structure of the treatment sessions.

Difficulties with the Pace and Rigidity of the Expert Criteria. The second broad theme was difficulties with the pace and rigidity of the expert criteria. Parents reported feeling “defeated and frustrated” by the slow pace and the length of treatment, stating “it’s slow, slow to see results, slow to get through each part”. Several parents also stated that it was “hard to be patient in CDI while waiting to get to the disciplinary portion”. Relatedly, parents also reported feeling frustrated about meeting expert criteria. They reported that each week they would get closer to meeting the goal, and yet could not move on because of the “rigidity” within PCIT. 

Concerns About Fit of PCIT. The third broad theme indicated that parents had concerns about whether PCIT was the right fit for their child and family. For example, even though all the children were within the PCIT age limit, several parents stated that their child was “too old to play” and “didn’t play anymore” and that they had struggled to continue with PCIT given the play-based structure. Relatedly, parents also stated concerns about how the structure of PCIT would help them with their day-to-day struggles in the home. For example, parents stated that when they had a difficult day or week with their child and were instructed to have special time, they felt that the structure of PCIT was not helping them manage disruptive behaviors more effectively at home.

Challenges with PDI. The fourth broad theme was related to challenges with PDI. Several parents stated that PDI was the most difficult part of PCIT. One parent reported that PDI felt like “CIA training” and that neither them nor their child appreciated it. Several parents also reported feeling guilty about putting their child in the timeout chair. Several parents added that they felt even more guilty when actively ignoring their child in the backup room, as they worried that their child felt “trapped and unsafe”. Some parents also reported that they disagreed with the approach of providing direct commands for the “sake of creating conflict”. One parent reported feeling “bothered” when they were instructed to wait it out and ignore requests from their child asking to go the bathroom during timeout. Lastly, many parents stated that PDI was “nearly impossible” to follow through at home on their own without coaching, leading them to feel increasingly helpless and hopeless throughout the process. 

Difficulties due to COVID-19 Pandemic. The fifth broad theme was related to the onset of the COVID-19 pandemic, which they reported interfered with their ability to engage in treatment. Several parents reported that once the treatment went virtual, it was harder to complete. Other parents reported that things “got too crazy” during the start of the pandemic and it was too difficult to have “another thing to do at that crazy time”.

Other Factors. The final broad theme was related to other factors that made PCIT difficult to complete. One factor was the stress PCIT placed on the parent. Parents stated that PCIT was “a big responsibility” and that “the resistance” children demonstrated to special time took “such a toll” on parents and that “there were some really tough moments”. Another factor that parents highlighted was the stress created by disagreements between parents about PCIT, including a lack of buy-in and a discrepancy in skill use, which impacted the parents’ ability to move on to PDI at the same time. One parent stated, “Not only did I have homework and I have to try to do special time and everything, but like I had to try to get my husband to do it. He wasn’t into the treatment rules”. Still another factor that made it difficult was when parents had issues with the clinician providing PCIT. For example, one parent reported that their child who was looking forward to the session went “into a rage” when their therapist had to cancel at the last minute, which contributed to them ultimately dropping out. 

## 4. Discussion

The purpose of the current study was to learn about the experiences of parents who drop out from PCIT before meeting graduation criteria. Our analysis yielded eight main themes that were incorporated into three main theoretical constructs: positive experiences in CDI that no longer necessitated treatment difficulties and challenges with PCIT, and the need for more clarity and orientation upfront. The existing literature and treatment strategies were applied to the data to facilitate our understanding of the parents’ experiences and to suggest remediation strategies. 

The recent literature has reconceptualized dropout from PCIT. In 2019, Lieneman et al. [38] published a paper regarding the impact of PCIT on child misbehavior after as few as four sessions. Their results suggest that significant improvements in children’s behavior can be experienced even by families who terminate early from PCIT [38]. Although historically, families who leave PCIT before graduation are considered as treatment failures, these results indicate that even small doses of PCIT can be associated with significant improvements in functioning. The results of the present qualitative study further corroborate Lieneman et al.’s [38] findings, as some parents reported that their issues were resolved after completing CDI, as evidenced by a reduction in ECBI scores, and that they did not need to continue with PCIT. Thus, it is possible that PCIT in small doses could have a positive impact and that dropout may not necessarily equate with treatment failure. 

Furthermore, the dropout rate in the present study decreased following the transition to internet-based PCIT, from 32.5% to 32.3%. Of the five families that dropped out of treatment post-March 2020, four of them cited COVID-19 as the reason. It is possible that the transition to internet-based PCIT may have helped retain other families who may otherwise have cited logistical barriers or a belief that PCIT felt “artificial” in a clinic as reasons for dropping out of treatment, as therapists were able to provide coaching to parents while managing their children in their own homes.

Though there were families who did not complete treatment because they felt PCIT had adequately helped their child, there were other families who felt that PCIT could be improved to help retain them in treatment. We believe that there are several existing strategies that may enhance PCIT and help reduce attrition rates. 

First, it may be beneficial to continue orienting clients to the treatment throughout. PCIT and cognitive and behavioral treatments generally begin with a psychoeducation and orientation phase of treatment, during which the client is presented with the rationale for the treatment and interventions. In addition, other cognitive and behavioral treatments provide psychoeducation throughout, including explicit instructions and directions on how to participate in therapy tasks and the rationale for the intervention, which is clearly linked back to the client’s goals. Furthermore, the client is oriented to the intervention being proposed, why it is being proposed, and how to do it. This is not generally the case in PCIT and may be important to incorporate, especially given the emotional challenges parents face in enacting new behaviors to manage children with disruptive disorders. 

Such repeated orientations are referred to as micro-orientation in DBT, which clinicians frequently use throughout treatment to work with emotionally dysregulated clients. DBT is a third-wave intervention that combines CBT, mindfulness, and dialectical philosophy. Because change interventions can be experienced as invalidating and/or difficult for such clients, the change focus often evokes emotion dysregulation that may disrupt collaborative work on therapy tasks [37]. Consequently, DBT therapists frequently explain why a particular treatment task is necessary to reach the client’s goals and need to instruct the client specifically on how to complete the therapy task despite or in the face of emotion dysregulation. 

Parents engaging in PCIT may themselves become dysregulated while using new skills to manage their child’s intense disruptive behaviors, particularly during PDI. Hence, providing repeated micro-orientations during PDI, specifically for the timeout procedure, may help to remind parents why they are being instructed to follow the PDI procedure. For example, at the start of a PDI session, micro-orienting could involve briefly reminding parents of the command-timeout sequence and the rationale behind this by saying something such as “it’s not about teaching your child to stack the blocks, we’re teaching your child to listen once you give a direct command”. Parents who become dysregulated by following through with the PDI procedure itself may be reminded that “while this procedure may feel difficult at first, it’s important to teach your child that you will be consistent and predictable no matter what. That helps your child feel safe within the boundaries you are setting”. While coaching provides the instructions and reminders of the logistics of the PDI plan, micro-orienting would focus more on reminding parents of the *why*, namely, why they should be following through with the timeout plan and how it connects to their longer-term goals. This is similar to the educational level of coaching outlined in Cheryl McNeil’s draft Coding CDI Coaching form [39]. McNeil [39] describes the educational level of coaching as providing education on “(1) the effect a skill could have on behavior in the future and (2) pointing out an immediate effect of the skill on child behavior” (p. 1). This level of coaching is encouraged in PCIT, and so formally including micro-orientation to PDI may help with parents who feel that they do not understand the rationale for PDI and why it is important. 

Motivational interviewing (MI) may be another strategy that could be incorporated into PCIT to reduce treatment dropout. Motivational interviewing (MI) is an evidence-based approach to treatment that involves engaging clients in identifying and committing to change-focused behaviors while resolving ambivalence [40]. The primary strategies used in MI include open-ended questions, affirmations, reflections, and summaries (OARS) to help resolve ambivalence and enhance motivation for change [40]. MI also involves empathizing with clients, rolling with resistance, and highlighting the discrepancies between stated goals and current behaviors in a non-confrontational and validating manner. Parents often feel that the expectations and skills practiced in PCIT are difficult and time-consuming, which can make it more likely that they drop out of treatment. A PCIT therapist using MI strategies throughout treatment may help a parent feel less judged and more motivated, and may facilitate a more supportive and collaborative atmosphere. Additionally, using MI to remind parents of their goals may enhance their commitment to follow through with HW, special time practice, and persevere through difficult moments in the treatment. 

Through a series of vignettes, N’zi et al. [41] describe how PCIT therapists can incorporate MI strategies to enhance parental motivation and decrease ambivalence in treatment. They suggest that MI can be used within PCIT to address initial ambivalence about beginning PCIT, improve homework adherence, address common barriers in the CDI and PDI phases of treatment, improve parental self-efficacy, and reduce risk of attrition. However, this study did not implement or analyze MI-based suggestions. Another study [42] designed to reduce premature termination from traditional PCIT using motivation enhancement sessions found that parents who received motivational enhancement reported increased willingness to change their parenting behavior. However, unfortunately, the study found no difference in dropout rates between those in traditional PCIT versus those in the motivational enhancement PCIT group [42].

Other researchers have directly studied the impact of a motivational interviewing intervention on parental engagement and treatment retention in PCIT among child welfare patients with varying levels of motivation. Chaffin et al. [27] examined the effects of an adjunctive motivational intervention across two parenting programs: PCIT and a standard didactic parenting training group. Parents in the study were randomly assigned to either a six-session self-motivation orientation group (based on motivational interviewing principles) or a six-session standard informational group, and parents who completed the pre-parenting sessions were then randomly assigned again to PCIT or the standard didactic parenting training group. The results of this study found that parents with low-to-moderate motivation at baseline who received the motivational pretreatment in combination with PCIT significantly increased their likelihood of remaining in treatment [27]. However, the study found no effect or negative effects for parents whose initial motivation was relatively high. Interestingly, the results found improvements in retention following the self-motivation group in those who completed PCIT but not for those in the standard didactic parent training group, suggesting a favorable interaction between the two rather than any single factor accounting for the improvements [27]. In other words, a pretreatment motivational enhancement intervention may be a good option for families who demonstrate low motivation or ambivalence, as the increased readiness for change following the MI-based intervention prepares them for the action and change-focused orientation of PCIT. 

Based on our findings, a third strategy may be drawn from acceptance and commitment therapy to improve treatment retention in PCIT. Specifically, addressing wavering commitment to PCIT by connecting parents’ stated goals with their values may be helpful in retaining families in treatment. Acceptance and commitment therapy (ACT) is an evidence-based treatment that encourages compassionate acceptance of our own experiences and emotions, rather than engaging in avoidance strategies, in order to pursue a life better in line with our values. ACT centers around psychological flexibility, or the ability to be fully present in the moment without judgment, defensiveness, or reactivity, and to change or persist in behaviors that are in line with one’s values [43]. Within ACT, psychological flexibility contains six interrelated skills: acceptance, cognitive defusion, contact with the present moment, self-as-context, values, and committed action. Several studies have explored the use of ACT with parents in different ways, including addressing parent psychological well-being, augmenting existing behavioral parenting interventions, and as a parenting intervention to directly address child behaviors or parenting difficulties [44]. ACT has shown promise as an intervention in helping parents manage stress and difficulties in relation to children with chronic illnesses, autism, and physical health needs [45].

Another key component of ACT involves mindfulness of the present moment and reducing experiential avoidance. Experiential avoidance is any attempt to escape, avoid, or suppress any unwanted internal or external experiences [46]. As PCIT can often lead to emotional dysregulation for both parent and child, especially in the PDI phase, parents might engage in experiential avoidance in several ways: not completing the ECBI, not completing CDI or PDI homework, avoiding giving commands or following through with timeout warnings, canceling appointments, and even dropping out of treatment. Rather than blaming or shaming parents for engaging in experiential avoidance, a compassionate PCIT clinician could incorporate concepts of ACT by responding to parents nonjudgmentally, helping parents make room for their difficult emotions or thoughts during the parent check-in or check-out at each session by cultivating mindfulness and psychological flexibility, and assisting parents with commitment to engaging in difficult parenting behaviors that are in line with their values. For example, several parents reported feeling frustrated with the pace of treatment and the slow progress. Principles of ACT to respond to these parents could involve modeling willingness to tolerate the parent’s frustration while validating the parent’s experience, and acknowledging how difficult it can be to follow through with a challenging treatment such as PCIT, while also exploring whether they are willing to tolerate their painful feelings if it means they are acting in line with their values to be a more consistent, warm, and authoritative parent. Coyne and Murrell [47] describe specific ways that ACT-based strategies can be applied while simultaneously practicing PCIT-based activities. One such example encourages parents to practice mindfulness of the present moment while engaging their child in special time, paying attention to the child’s needs rather than reacting to their own internal worries [47]. Another example includes helping parents plan and take actions in difficult parenting moments that are in line with their parenting values [47]. Helping parents connect with their long-term goals and values, and then to behave in ways that are in line with those values, lends itself easily to PCIT as coaches can facilitate willingness to engage in treatment by reminding parents of their values throughout treatment.

While there is certainly room for growth in PCIT to help address these reasons for dropout, recent adaptations to PCIT, if used consistently, could also help reduce dropout. For example, one reason identified for dropping out of treatment is that PCIT did not help parents directly manage disruptive behaviors at home during CDI. Parents often felt that while special time was helpful and important, they struggled to manage aggression or non-compliance prior to beginning PDI. In 2020, Cheryl McNeil created a resource for parents called the “Cooperation Chart” to help them manage disruptive behaviors at home during the COVID-19 pandemic. The Cooperation Chart is a tool parents and caregivers can use to help communicate with their children about their behaviors, using praise and warnings to identify positive and negative actions, respectively. Parents and caregivers are encouraged to monitor their child’s behavior by identifying both positive and negative actions and tallying them on the chart. If a child receives mostly happy faces at the end of the time period, they receive a surprise reward [48]. We believe the Cooperation Chart is a tool that could be used during CDI to help manage children’s disruptive behavior without using commands or timeout and is in line with the spirit of CDI. If used in tandem with CDI, we believe this could help parents manage disruptive behavior effectively until they reach PDI and can use commands and timeouts as needed. This may help prevent PCIT dropout by giving parents more tools to manage behavior during CDI.

Parents also reported that the rigidity and stringency of the coding and expert criteria contributed to their dropout. As such, it is possible that a more flexible approach may be used to determine if a family can move on to PDI when there is risk for attrition. In an adaptation of PCIT for traumatized children (PCIT-TC), there is a “readiness for PDI checklist” that helps clinicians determine whether or not to move caregivers on to PDI [49]. The first item on this checklist asks whether the family is at risk of dropping out. The criteria used to determine this risk are: client is at CDI 9 or above, parent seems defeated and hopeless about achieving mastery or completing PCIT, and/or parent is canceling on a regular basis (e.g., two out of four sessions per month). There are also other criteria to consider when deciding to move someone to PDI before meeting expert criteria in CDI. Considering dropout risk is most relevant to the present study and highlights the importance of considering parents’ own feelings of frustration and hopelessness with the treatment when making this decision. Using this sheet to guide a flexible approach to the CDI-PDI flow may help reduce PCIT dropout rates.

One of the aspects of PCIT that makes it unique and more effective than other parent management treatments is the emphasis on live or in vivo coaching for parents by the therapist to shape parents’ acquisition of specific parenting skills. However, not all coaching statements are created equally. It is possible that the style and type of coaching may influence not only the development of parental skills, but also the therapeutic relationship and the likelihood of attrition. In their initial evaluation of the Therapist–Parent Interaction Coding System (TPICS), which allows for direct coding of therapists’ coaching in PCIT, Barnett et al. [50] found that therapists use a range of both directive and responsive coaching statements. In looking at session-to-session change, Barnett et al. [50] found that responsive coaching mediated parents’ skills acquisition, with parents using more labeled praises in the following session, while directive coaching techniques were associated with fewer labeled praises and behavior description in the following session. In a follow up to this study, Barnett et al. [25] found that responsive coaching predicted faster completion of the CDI phase, while families who dropped out of treatment received fewer responsive coaching statements and more skill drills. It is possible that parents may perceive directive coaching and skill drills as being less validating and encouraging than responsive coaching. As suggested by this study, one way to buffer against dropout could be to incorporate a formal coach-coding system, such as the TPICS, in clinical practice in order to effectively evaluate the type of coaching statements used. This would influence not just the clinical treatment being delivered, but also the quality of training provided to therapists learning and delivering PCIT.

A common theme among many of the therapies and interventions described above is the idea of expressing empathy and fully validating parents’ concerns. While PCIT is a strong evidence-based treatment, with some time dedicated each session to discuss parents’ concerns unrelated to their child’s behavior, the reality is that many parents may need even more validation and compassionate guidance throughout treatment to feel fully supported. In the spirit of flexibility within fidelity, coaches could be ready to follow the manual while keeping each family’s individual needs and goals in mind, including being willing to fully assess and validate parents’ concerns before moving on to coding and coaching skills. Empathic listening and validation, coupled with sensitive and responsive coaching guided by short- and long-term parenting goals and values, may help reduce dropout rates and help more parents find greater harmony in their homes.

### Limitations and Future Directions

There are several limitations within our study. The sample was composed of mostly white, middle-to-upper-class parents, who were parenting with a stable partner. It is possible that the themes identified and conclusions made would have looked different had the sample been more diverse. However, it is worth noting that a majority of prior studies on PCIT were conducted on clients of lower SES. Another limitation might be that the impact of COVID-19 on sampling, adherence, outcomes, etc., is largely unknown, depending on when the COVID-19 pandemic occurred in the timeline of the study. It is possible that the results may have been different if the data had been collected completely prior to COVID-19.

Several themes were identified that could potentially reduce dropout and increase the impact of PCIT. Future research could incorporate specific suggestions made by parents into standard PCIT and assess their impact on dropout. Based on the theoretical constructs identified in our study, future studies could also incorporate strategies such as repeated micro-orientation, motivational enhancement, acceptance, and value-driven long-term goals into standard PCIT and evaluate their impact on dropout rates. Another direction for future research could be to combine the quantitative analyses of dropout with a qualitative analysis that explores parents’ perceptions of these adaptations to treatment. This mixed-methods approach could illuminate whether these changes actually address identified parental concerns.

## 5. Conclusions

PCIT is considered one of the strongest evidence-based treatments for young children aged 2–7. However, dropout rates are high. The present study explored parental perceptions about PCIT and reasons for dropout. Results highlight various themes that were identified and adaptations that could be made to PCIT to combat these issues. If results can impact PCIT, then hopefully more families will be able to remain in PCIT, which will reduce the prevalence of disruptive behavior disorders in children.

## Figures and Tables

**Figure 1 ijerph-19-14341-f001:**
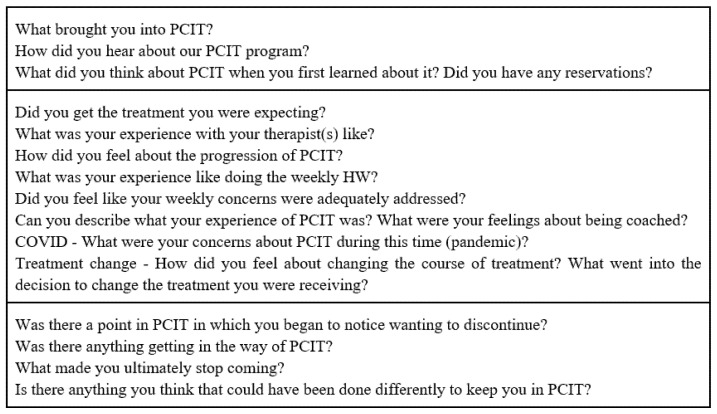
Interview questions.

**Figure 2 ijerph-19-14341-f002:**
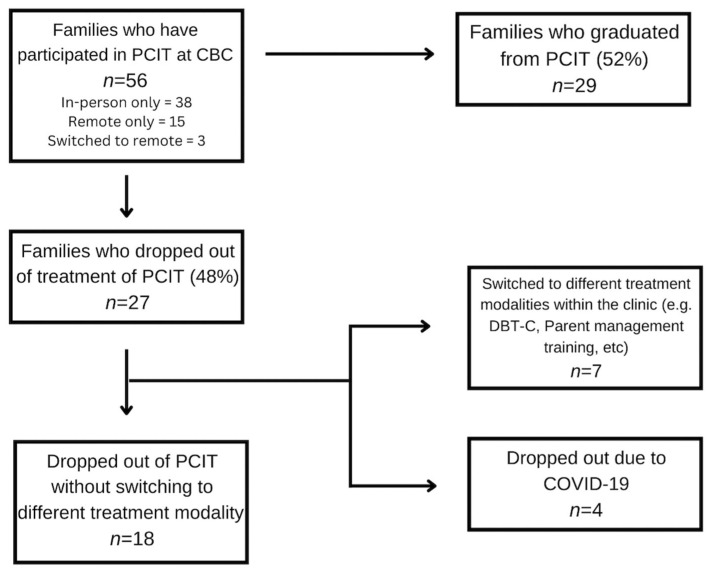
Flow Chart Describing Dropout.

**Table 1 ijerph-19-14341-t001:** Demographic Information of Sample.

Demographic Information of Sample		
Variable	M	SD
ECBI Intensity Pretreatment Score	147.11	24.59
ECBI Problem Pretreatment Score	17.11	7.59
ECBI Intensity at Dropout Score	120.57 *	41/73
ECBI Problem at Dropout Score	12.29	10.19
Age of Child	6.42	0.79
Number of Sessions Attended	13.43	7.93
Variable	*n*	%
Parental Engagement in PCIT		
1 Parent in PCIT	2	28.57%
2 Parents in PCIT	5	71.43%
Annual Income		
Less than USD 50,000	1	14.29%
USD 70,000–80,000	1	14.29%
USD 300,000 or more	5	71.43%
Number of Children in Family at Time of Treatment		
1	1	14.29%
2	3	42.86%
3	3	42.86%
Diagnosis of Child		
ADHD	1	14.29%
ODD	3	42.86%
Unspecified Disruptive Behavior	2	28.58%
Separation Anxiety Disorder	1	14.29%
Generalized Anxiety Disorder	1	14.29%
Unspecified Anxiety Disorder	1	14.29%

* Below clinical cutoff of 131 for ECBI.

## Data Availability

Not applicable.

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
