# Peer review of "A Qualitative Analysis of Attrition in Parent–Child Interaction Therapy"

_ijerph, 2022, doi:10.3390/ijerph192114341_

Round 1

Reviewer 1 Report

I enjoyed the opportunity to review this submission. It is a well-written article and will be of relevance and interested to clinicians. I particularly appreciated the discussion in which authors provided novel applications and strategies for reducing attrition in PCIT. Good work! 

1.      Original Submission

1.1  Recommendation:

Revise and Resubmit

2.      Comments to Author:

MS. Ref. No.: ijerph-1917815

Title: A Qualitative Analysis of Attrition in Parent-Child Interaction Therapy

Overview and general recommendation:

Parent-Child Interaction Therapy (PCIT) is consistently classified as a best-practice for children and families with histories of disruptive behavior disorders and other externalizing behavior problems. Although it has a strong evidence-base for reducing problematic behaviors in children and improving the caregiver-child relationship, it is plagued with high dropout rates, like many other behavioral parent training interventions. The current study investigated reasons for premature termination of treatment using a qualitative approach.

The current study is on a topic of relevance, interest, and importance to the readers of this journal. I look forward to seeing it in print after minor issues are addressed. See below for recommendations.

2.1  Comments:

1.      Introduction (page 2): The authors offered a well-written and concise description of the two phases of treatment and of the goal criteria for PCIT. Rather than referencing the “expertise in specific parenting skills” the authors could consider referencing the identified “goal criteria” of skills in both CDI and PDI.

2.      Introduction (Page 2): Regarding graduation criteria, the authors state that the criteria can be “difficult for families to meet.” The authors could consider revising this sentence. As it reads, it sounds like the goal criteria are too difficult, suggesting that many caregivers cannot achieve this. Data would argue that many families are successful at PCIT and in achieving the goal criteria. It is my believe that the sentence could stand without the last comment (i.e., “Strict graduation criteria are in place to ensure that the greatest therapeutic gains are achieved.” Particularly since you introduce similar content in the third paragraph on this page.

3.      Introduction (Page 2): The word “the” appears to be missing from the following sentence: “Research shows that treatment gains generalize to the treated children’s school behavior and to….”

4.      Introduction (Page 2, paragraph 3): The authors cite research that “fewer child MDD diagnoses” are associated with premature termination from PCIT. However, the articles cited with this statement do not indicate that any child factors, particularly MDD diagnoses, had an association with outcome. Werba, Eyberg, Boggs, and Algina (2006) note “child variables demonstrated little predictive value for PCIT outcome.” The second article cited (Niec et al., 2017), examines therapist responsivity and directivity as it related to treatment completion. Thus, neither of the articles cited support that childhood variables, particularly MDD status, are implicated in treatment completion.

5.      Participants (Page 3): Why did the authors include “at least one session of PCIT” as the minimum criteria? Do the authors believe that participants who attended one session might be different from participants who attended 5 or 10 sessions? Might their experiences or reasons for discontinuing be different? The authors could provide some rationale as to why a minimum of one session was required as opposed to higher. The authors could also consider including data on how average number of sessions participated for families included in the study.

6.      Procedure (Page 4): The authors could consider describing the use of convenience and theoretical sampling in obtaining the final sample.

7.      Data Analysis (Page 5): The authors provide a helpful and thorough description of data analysis process. Did the authors use open, axial, and/or selective coding? If so, it would be helpful to describe that process and how interviews were coded. How was memoing used in data analysis? What did the authors to do ensure study rigor per Strauss and Corbin guidelines on developing grounded theory?

8.      Results (page 6): The authors provide a very helpful flow chart of drop out. Of those that dropped, how many were receiving internet-based treatment? Did this include families who may have started treatment in-person? More detail could be helpful around this.

9.      Results (Page 7): The authors did a nice job of organizing the themes/theoretical constructs. Regarding Construct 1, I am curious as to the average number of sessions attended for families who reported they no longer needed treatment, particularly given the authors reports that families were only required to attend one session.

10.  Page 8: The authors do a nice job of giving voice to the participants in the study. Powerful quotes were selected to underscore themes.

11.  Discussion (Page 9): The authors do a nice job of suggesting strategies for enhancing PCIT and reducing attrition. The author’s argument that generally in PCIT therapists do not orient the client to the intervention, why it is being proposed, and how to do it, is arguably inaccurate. As a PCIT clinician, I often provide rationale for skill use and tie skills back to client’s identified goals (both during in vivo coaching and in pre-and post-coach). I would argue that many clinicians do this. The authors might consider rewording to state that “This might not always be the case in PCIT” or something similar.

12.  Discussion (Page 10): I like the idea of labeling the coaching strategy of “micro-orientations” in coaching. This is a very helpful tool for skill acquisition.

13.  Discussion (Page 11): The use of ACT strategies is a novel application for PCIT. This is well-articulated by the authors. I believe it would be a good fit for PCIT families.

14.  Discussion (Page 12): The authors mention the Cooperation Chart as developed by Cheryl McNeil. Have there been any published studies documenting the use of the Cooperation Chart as an adjunct to PCIT treatment? Empirical data on this could be helpful.

15.  Discussion (Page 12; 3rd paragraph): PCIT International has largely moved away from the term “mastery” and has replaced it with “goal criteria.” While I am unclear if PCIT-TC has followed suit, it may be worth clarifying and changing, if so.

Reviewer 2 Report

Dear Authors,

The manuscript focus Parent-Child Interaction Therapy and how to avoid disruption in order to achieve improvements in a child's behavior, as a public health issue. The article is interesting, but I have some comments.

- Please identify the significance of all abbreviations in the abstract and in the main body (eg. SES, MDD, CDI).

- use neutral writing, delete words as fortunately and unfortunately.

- Fernandez and Eyberg are an example of the few interventions that have been implemented, but since their publication in 2009, has anything else been conducted? The introduction would benefit from more literature and more recent ones. Namely, those that you felt necessary to explore in the discussion part.

- Regarding Materials and methods, what is the dimension of the universe? What is the period of time that you collected the empirical data? Could you provide more detail about the interviews, perhaps examples of the questions or their topics? Please identify the transcription software. The empirical study is inspired by Auerbach and Silverstein, but I suggest stating clearly that Glaser and Strauss are (recognised as) the founders of grounded theory. Figures must have a source, is it your own elaboration (?), so state it clearly.

- you refer to the transition to internet-based PCIT, what are the implications of the pandemic and the at-distance therapy? More attention and detail should be paid to the dropout rate decreased.

- the motivation for and expectation about the therapy are relevant for keeping in the program, as well as the person who detected the need and who referred the parent-child. 

- Results: please include proper quotes, they are expected to be rich for interpreting the empirical results (identifying the participants within codes). construct one has 8 lines and construct three has 1,5 pages, I suggest and homogenous approach. 

- Discussion: I found it a bit confusing. I suggest it be reorganized. A discussion should present the relation between your results and previous studies, instead of new information (ex. motivation/ MI). Please add it to the introduction and in the results. 

- References: Please, search for more recent literature, since many references are more than 20 years old.

I wish you success.
